# Effects of Simultaneous Exposure to a Western Diet and Wheel-Running Training on Brain Energy Metabolism in Female Rats

**DOI:** 10.3390/nu13124242

**Published:** 2021-11-26

**Authors:** Marta Maria Nowacka-Chmielewska, Daniela Liśkiewicz, Konstancja Grabowska, Arkadiusz Liśkiewicz, Łukasz Marczak, Anna Wojakowska, Natalia Pondel, Mateusz Grabowski, Jarosław Jerzy Barski, Andrzej Małecki

**Affiliations:** 1Laboratory of Molecular Biology, Institute of Physiotherapy and Health Sciences, Academy of Physical Education, 40-065 Katowice, Poland; d.liskiewicz@awf.katowice.pl (D.L.); natalia.pondel1@gmail.com (N.P.); a.malecki@awf.katowice.pl (A.M.); 2Department for Experimental Medicine, Faculty of Medical Sciences in Katowice, Medical University of Silesia, 40-055 Katowice, Poland; grabowska.i.konstancja@gmail.com (K.G.); mateusz.m.grabowski@gmail.com (M.G.); jbarski@sum.edu.pl (J.J.B.); 3Department of Physiology, Faculty of Medical Sciences in Katowice, Medical University of Silesia, 40-055 Katowice, Poland; adliskiewicz@gmail.com; 4Institute of Bioorganic Chemistry, Polish Academy of Sciences, 61-704 Poznań, Poland; lukasmar@ibch.poznan.pl (Ł.M.); astasz@ibch.poznan.pl (A.W.)

**Keywords:** western diet, wheel-running training, female rats, global brain proteome, brain energy metabolism

## Abstract

Background: In the pathogenesis of central nervous system disorders (e.g., neurodegenerative), an important role is attributed to an unhealthy lifestyle affecting brain energy metabolism. Physical activity in the prevention and treatment of lifestyle-related diseases is getting increasing attention. Methods: We performed a series of assessments in adult female Long Evans rats subjected to 6 weeks of Western diet feeding and wheel-running training. A control group of lean rats was fed with a standard diet. In all experimental groups, we measured physiological parameters (animal weights, body composition, serum metabolic parameters). We assessed the impact of simultaneous exposure to a Western diet and wheel-running on the cerebrocortical protein expression (global proteomic profiling), and in the second part of the experiment, we measured the cortical levels of protein related to brain metabolism (Western blot). Results: Western diet led to an obese phenotype and induced changes in the serum metabolic parameters. Wheel-running did not reduce animal weights or fat mass but significantly decreased serum glucose level. The global proteome analysis revealed that the altered proteins were functionally annotated as they were involved mostly in metabolic pathways. Western blot analysis showed the downregulation of the mitochondrial protein—Acyl-CoA dehydrogenase family member 9, hexokinase 1 (HK1)—enzyme involved in principal glucose metabolism pathways and monocarboxylate transporter 2 (MCT2). Wheel-running reversed this decline in the cortical levels of HK1 and MCT2. Conclusion: The cerebrocortical proteome is affected by a combination of physical activity and Western diet in female rats. An analysis of the cortical proteins involved in brain energy metabolism provides a valuable basis for the deeper investigation of changes in the brain structure and function induced by simultaneous exposure to a Western diet and physical activity.

## 1. Introduction

The Western diet pattern characterized by a high daily intake of saturated fats and refined carbohydrates often leads to overweight and obesity, which increase the risk of several debilitating and deadly diseases, including diabetes, heart disease, and some cancers [1]. In addition, long-term exposure to highly palatable foods typical of a Western diet has been linked to cognitive impairments in both animal models and humans [2,3]. Western diet-induced adverse effects in the brain seem to be related to disturbances of brain energy metabolism, especially to the principal glucose metabolic pathways. The disturbances of glucose homeostasis can be due to the well-recognized development of insulin resistance, but they may also be caused by impaired glucose transport into the brain [4]. Glucose is reversibly transported from arterial blood across the membranes of endothelial cells into and from brain cells via isoforms of equilibrative glucose transporters (GLUTs) that have different kinetic properties, with GLUT1 localized mostly in the endothelium and astrocytes and GLUT3 and GLUT8 localized mostly in neurons [5]. Thus far, it has been merely shown that such a diet reduced GLUT1 expression in the hippocampus [6]. A Western diet may also induce changes in the levels of other compounds important for brain energy homeostasis [7]. It has been observed that a Western diet lowered the hippocampal expression of the MCT1 transporter, which is responsible for moving monocarboxylates across the blood–brain barrier [6].

Physical activity has been associated with a plethora of functional, cellular, and molecular alterations within the brain [8,9]. Regular exercise correlates with an increase in the amount of neurotrophic factors and markers of synaptic plasticity [10], reduces inflammatory factors [11], as well as contributes to the improvement of mood and cognitive abilities (including memory and learning) [12]. Studies published thus far show that the energy challenge caused by exercise can affect the CNS by improving cellular bioenergetics, stimulating the removal of damaged organelles and molecules, and attenuating inflammation processes [13,14]. However, little is known about why it has such a profound effect on the brain. Neurobiological mechanisms associated with physical activity are not entirely known, which is partly due to a lack of uniformity and parameterization in experimental protocols employed to assess the impact of exercise in animal models [15]. Therefore, in the present study, we have used a forced wheel-running protocol in which the same training load is applied to all the subjects [15,16].

Our primary goal was to investigate the effects of simultaneous training (wheel-running) and a Western diet on the cerebrocortical proteome profile. In the second part of the experiment, we verify the levels of particular proteins related to brain energy metabolism in the frontal cortex. Both the frontal and temporal lobes have some common functions, including working and long-term memory and emotional functions [17,18]. These structures are also of particular importance to recent clinical and animal studies related to the beneficial effects of physical activity on the brain [19,20,21].

Finally, pervasive sex differences in metabolic traits, such as body fat distribution, glucose homeostasis, insulin signaling, ectopic fat accumulation, and lipid metabolism have often been omitted in human and animal model research [22]. Considering that susceptibility to mental and metabolic diseases is strongly associated with sex [23] and exposure to environmental factors such as unhealthy diet and a lack of physical activity [24], we decided to perform our study on the female rats.

## 2. Materials and Methods

### 2.1. Animals and Experimental Groups

All animals were provided by the Animal House of the Department of Experimental Medicine, Medical University of Silesia (Katowice, Poland) and were treated in accordance with Directive 2010/63/EU for animal experiments using the protocols approved and monitored by the Local Ethics Committee for Animal Experimentation in Katowice (protocol number 62/2016). Sample sizes that were used in the present study were chosen based on our experience and other investigators conducting similar experiments. The minimum number of rats required to obtain consistent data was used, and every effort was taken to minimize the suffering of the animals. Before the study began, the rats were adapted to their new conditions for one week. Rats were housed 3–4 per cage in a climate-controlled room (22 ± 2 °C, relative humidity: 55 ± 10%) with a 12 h:12 h light/dark cycle starting at 07:00 a.m., and they received food and water ad libitum.

The rats were randomly divided into 3 experimental groups. All interventions (diet and wheel-running procedure) lasted for 6 weeks. Upon the initiation of the experiments, the rats (average weight 220 g, 9–10 weeks old) were either maintained on a standard diet as controls (CTR group, *n* = 12) or switched to a Western diet with six human snack foods varied daily in addition to standard chow (WD group, *n* = 12). The animals in group WD/EX (*n* = 11) were subjected to the exercise (wheel-running) procedure and fed with a Western diet.

### 2.2. Dietary Protocol

The standard diet (standard rodent chow) energy content (3.57 kcal/g) comes from 67% carbohydrates, 25% proteins, and 8% fats. To mimic the human obesogenic diet, animals were fed with commercially available human snacks as previously described [25]. The animals received one of two sets of snacks interchangeably. These were given to the animals on alternate days as one diversified diet. Set 1 included the following: candy bar (Mars; Mars Inc., McLean, VA, USA), crackers (Lajkonik Snacks, Skawina, Poland), and kabanos (dry sausage made of pork; Tarczyński, Trzebnica, Poland). The following snacks were in set 2: candy bar (Bounty; Mars Inc., McLean, VA, USA), potato chips (Lays Salt; PepsiCo, NC, USA), and Tilsit cheese (Hochland SE, Heimenkirch, Germany). WD rats received clean water and a sweet beverage—10% fructose solution (39.8 kcal/100 mL; sweetened with Consweet Sweetener Confex-Product, Warsaw, Poland) in a second container. The average caloric density of these two dietary sets (including 10% fructose) was 4.84 kcal/g with the following caloric profile: carbohydrates 33.2%, fat 33.1%, and proteins 16.6%. In each group, all foods were provided ad libitum. All animals had free access to water for the duration of the experiment.

The food was supplied daily, and food intake was monitored every day (chow and snacks were weighed before and after consumption). Liquid intake (water and 10% fructose) were monitored every second day. The diet intake in each cage was calculated as Wd  =  initial food weight—(leftover food weight + spilled food weight). Total energy intake was determined by calculating the combined intake of liquid fructose, snacks, and chow. The daily energy intake per rat (kcal/day/rat) was calculated as: Wd/7/*n* * Et (Et is total energy of a snacks and standard chow), *n* is the number of rats in the cage.

Composition and nutritional profile of the Western diet is provided in the Appendix A.

### 2.3. Wheel-Running Training

In the present study, six custom-made polycarbonate motor running wheels were used. The advantage of this system is the possibility of controlling the training; all animals run at the same speed for the same period of time, in contrast to voluntary activity where the level of activity depends on the animal’s preferences [15]. The rats were exercised 5 days/week for 6 weeks, of which the first week was an introduction week (the habituation phase). During the habituation phase, both intensity (speed) and volume (time) were increased following upward progressive patterns, as summarized in Figure 1. For the next 5 weeks, the rats were subjected to the wheel-running incremental training, in which the animals reached a final speed of 15 m per minute and a total distance of 900 m daily (3 times for 20 min of wheel-running with 5-min rest). No electric shocks were used, so if the rat refused to run, it was motivated by a short pause. After this, if the rat persisted in its refusal to run, it was removed from the wheel and re-introduced to it at a later time. About a 75% success rate response was observed; that is, 11 rats from the group of 15 rats subjected to the wheel-running training protocol managed to complete the habituation phase.

### 2.4. Tissue Collection

Rats were fasted overnight before tissue collection. The following morning (at 7:00 a.m.), they were decapitated, and trunk blood was collected. In order to eliminate hormonal fluctuation, rats were sacrificed alternately (one animal from each group). The brain was rapidly removed and put on an ice-chilled metal plate, dorsal side up. Subsequently, approximately 2.0 mm-thick slices of the frontal and temporal cortices were removed from each hemisphere. The brain structures (frontal cortex for Western blotting analysis) and temporal cortex for liquid chromatography-tandem mass spectrometry (LC-MS/MS) analysis) were weighed and stored at −80 °C until usage. The blood samples were centrifuged at 2500× *g* for 10 min (4 °C). Then, the collected serum was stored at −80 °C for subsequent analysis.

The two components (abdominal fat mass and lean body mass) of the body composition were measured. The average content of the retroperitoneal and subcutaneous adipose tissue in the animals was assessed by weighing. After the removal of the internal organs, fat, skin, and tail, the trunk of the decapitated animal was weighed and considered as a lean body mass.

### 2.5. Biochemical and Hormonal Assays

For the determination of biochemical and hormonal parameters, the rats were food deprived for 8 h before testing. Serum glucose levels were measured using a Mindray BS-200 Chemistry Analyzer (Shenzhen Mindray Bio-Medical Electronics Co., Shenzhen, China). The serum concentrations of insulin and leptin were determined using commercial ELISA kits (insulin: R&D Systems, Minneapolis, MN, USA; leptin: Thermo Fisher Scientific, USA; leptin: Thermo Fisher Scientific, Waltham, MA, USA) according to the manufacturer’s instructions. Serum 17-β estradiol level was measured with a chemiluminescent immunometric assay on an IMMULITE 2000 analyzer (Siemens Healthcare Diagnostics, Eschborn, Germany). For the insulin assay, the intra-assay coefficient of variation (%CV) was <10%, and the sensitivity was 5 μlU/mL. For leptin, the %CV was 4.3%, and the sensitivity of the assay was 22 pg/mL. For the 17-β estradiol assay, the %CV was respectively 4.5% and the sensitivity was 2 pg/mL.

### 2.6. Global Proteomic Profiling

The temporal cortices from groups: CTR (*n* = 11), WD (*n* = 12), and WD/EX (*n* = 7) were used for liquid chromatography-tandem mass spectrometry (LC-MS/MS) for protein analysis as previously described [25] (detailed in Appendix B). The PANTHER [26,27] and Metascape [28] functional annotation tools were used to identify the significant ontology associations. The complete list of *Rattus norvegicus* proteins detected by mass spectrometry-based proteomics was used for the background and the GO term subcategories “GOTERM_BP_DIRECT”, “GOTERM_CC_DIRECT”, and “GOTERM_MF_DIRECT” were selected for analysis. False discovery rate (FDR) correction of Fisher *p*-value was applied to identify the significant functional annotations (significance was considered when *p* < 0.05).

### 2.7. Western Blotting

For protein extraction, thawed tissue pieces (frontal cortices from groups: CTR, *n* = 6; WD, *n* = 6; WD/EX, *n* = 5) were homogenized by sonication in an RIPA buffer (300 µL for each 10 mg of tissue; Sigma-Aldrich Corp., MO, USA) with protease (Complete™ ULTRA Tablets, Roche Holding AG, Basel, Switzerland) and phosphatase (PhosSTOP™, Roche Holding AG) inhibitors and then centrifuged at 12,000× *g* for 20 min at 4 °C. The protein concentration was determined with BCA Reagent (Abcam, Cambrige, UK; ab102536). The BSA standard curve was used to calculate the total sample protein concentration. The samples containing 15 µg (GLUTs, MCTs), 20 µg (Acad9, Acat1, ATP5j), or 30 µg (HK1, G6PD) of total protein were separated on 4–15% or 8–16% SDS-PAGE precast gels containing fluorophore (Bio-Rad Laboratories, Inc., Hercules, CA, USA) and transferred onto PVDF membranes (Bio-Rad Laboratories, Inc.). Visualization of total protein (loading control) was performed directly in the exposed to UV light gel after electrophoresis. Membranes were blocked for 1 h at room temperature in Casein Blocking Buffer (Sigma-Aldrich Corp.) and incubated overnight at 4 °C (1× casein in deionized H_2_O) with the appropriate primary antibody (Appendix A summarizes the antibodies used in the experiment). After washing, the membranes were incubated with a secondary goat anti-rabbit IgG antibody (1:5000; Abcam, ab97051). Immunoblots were visualized by Clarity Western ECL Blotting Substrates (Bio-Rad Laboratories Inc.) and detected by the ChemiDoc™ Touch Imaging System (Bio-Rad Laboratories Inc.). The targeted proteins were quantified by ImageLab Software 5.2.1 (Bio-Rad Laboratories Inc.). After normalization to the loading total protein (stain free gels), the results were expressed as a mean ± standard deviation (SD) fold change of the tested samples in comparison to the control.

### 2.8. Statistical Analysis

Prism 9.01 (GraphPad Software, San Diego, CA, USA) was used for statistical analyses and figure generation. The results are expressed as the means ± standard error (SD). The distribution of each dataset was checked for normality using the Shapiro–Wilk test. Depending on the data distribution, either the one-way ANOVA with Tukey post hoc or Kruskal–Wallis with Dunnett post hoc were used for the estimation of significant differences between three groups. In the case of repeating measurements, two-way ANOVA with Tukey multiple comparison test was applied. In all analyses, *p*-values of less than 0.05 were considered to be statistically significant.

## 3. Results

### 3.1. Wheel Running Modifies the Biochemical, but Not Phenotypic, Changes Induced by the Consumption of a Western Diet

At baseline (week 0), the average weight of the rats was 204 ± 15 g, with no differences between experimental groups. Figure 2A shows mean body weight across the 6 weeks of the experiment. Two-way ANOVA analysis revealed that rats consuming a Western diet gained weight more excessively than controls (an intervention: F(2, 177) = 70.80, *p* < 0.0001), a time: (F(6177) = 35.2, *p* < 0.0001), and an interaction of these factors (F(12, 177) = 2.41, *p* = 0.0063), and showed significantly higher fat mass (ANOVA *p* < 0.0001, F(2, 32) = 25.52) and lean body mass (ANOVA *p* < 0.0001, F(2, 32) = 18.25), as compared to the control animals. Tukey’s post hoc test revealed that rats consuming a Western diet showed a greater body weight gain compared to the control rats (weeks 1–6: *p* < 0.01, *p* < 0.05 vs. WD group, *p* < 0.0001, *p* < 0.01, *p* < 0.05 vs. WD/EX group (Figure 2B,C). These observations were associated with changes in average energy intake. Namely, a significant increase in energy intake was observed both in the WD groups in comparison to the control (ANOVA *p* = 0.0005, F(2, 30) = 10.05, CTR vs. WD: *p* = 0.0007, CTR vs. WD/EX: *p* = 0.0039; Figure 2D). We observed that wheel-running training normalized a high level of glucose (Kruskal–Wallis test *p* = 0.0006, CTR vs. WD: *p* < 0.01, WD vs. WD/EX: *p* = 0.008, Figure 2E). The serum glucose levels were similar in the WD/EX and the CTR groups (304.0 vs. 307.1 mg/dL). In addition, exercise tended to decrease the Western diet-induced rise in the serum levels of leptin. Specifically, for the WD group, the serum leptin levels were by 3.5 and 2.7 times higher in comparison to the control (*p* = 0.022), and WD/EX (*p* = 0.04), respectively (Kruskal–Wallis test *p* = 0.0054, Figure 2F). The examination of the serum insulin levels showed no significant changes (ANOVA *p* = 0.49, F(2, 20) = 0.73, Figure 2G). To assess the potential influence of 17-β estradiol levels on the studied factors, we measured the serum level of this hormone at the end of the experiment. We did not observe statistically significant differences between the groups (Kruskal–Wallis test *p* = 0.47, Figure 2H).

### 3.2. The Combination of a Western Diet and Wheel Running Alters the Rat Cerebrocortical Proteome Profile

The analysis of proteomics data identified 909 proteins in the rat cerebrocortical samples. The analysis of LC-MS/MS data using the multiple-sample ANOVA analysis revealed that the levels of 80 proteins significantly differed between the studied groups (data presented as a heat map, Appendix A). The identified proteins are involved in many biological processes. The altered proteins were functionally annotated as factors involved in small molecule catabolic process, neuron maturation, and regulation of synaptic plasticity (Figure 3A). The identification information for these proteins, including accession number (IDs), gene, full protein name, fold change, and corresponding biological processes (GO:BP), are summarized in Appendix C.

Post hoc analysis showed that among the proteins that differed between the control and Western diet-fed animals, three were upregulated (Alpha-1-inhibitor 3—A1i3, Acetyl-CoA acetyltransferase, mitochondrial—Acat1, ATP synthase-coupling factor 6, mitochondrial—ATP5j) and four were downregulated (Acyl-CoA dehydrogenase family member 9—Acad9, Contactin-associated protein 1—Cntnap-1, Disks large homolog 4—Dlg4, Protein kinase C—PKC). Exposure to a Western diet and wheel-running resulted in decreased levels of 46 proteins and increased levels of 34 proteins. Among the 80 proteins common to the three group comparisons, one-fourth are known to be involved in metabolism pathways, especially mitochondrial. This is demonstrated by three representative examples in Figure 3B–D showing Acyl-CoA dehydrogenase family member 9 (Acad9) (ANOVA *p* = 0.014, F(2, 27) = 4.97), Acetyl-CoA acetyltransferase, mitochondrial (Acat1) (ANOVA *p* = 0.038, F(2, 27) = 3.69), and ATP synthase-coupling factor 6, mitochondrial (ATP5j) (ANOVA *p* = 0.009, F(2, 27) = 5.57).

Organizing the proteomic data by two interventions (WD and WD/EX) allowed us to determine which biological processes changed by a Western diet consumption were still affected after training (Figure 4A). There was an overlap between proteins regulated by a Western diet and by the wheel-running (Figure 4B,C). Western diet consumption altered 22 proteins compared to the CTR. In the WD/EX group, the abundance of 104 was altered. The identification information for these proteins, including gene and full protein name, are summarized in Appendix C.

### 3.3. Simultaneous Exposure to a Western Diet and Wheel Running Induced Changes in the Metabolic Proteins of the Frontal Cortex

In the second part of the experiment, Western blot analysis was performed for the three mitochondrial proteins that differed between the control and Western diet-fed animals: Acad9, Acat1, and ATP5j (Figure 5A). The cortical level of Acad9—a protein involved in fatty acid beta oxidation—had a 0.42-fold decrease in the WD (*p* = 0.0006) and 0.24-fold in the WD/EX groups (*p* < 0.0001) (ANOVA *p* < 0.0001, F(2, 14) = 21.43; Figure 5B). No changes were observed in cortical levels of Acat1 (Kruskal–Wallis test: *p* = 0.38; Figure 5C), the enzyme that catalyzes the reversible formation of acetoacetyl-CoA from two molecules of acetyl-CoA. In both groups fed with a Western diet (WD, WD/EX), a significant increase in the level of ATP5j was reported (ANOVA *p* = 0.02, F(2, 14) = 5.17 Figure 5D) as compared to the CTR. Namely, a 1.68-fold increase (*p* = 0.043) was observed in the WD, and in the WD/EX, a 1.75 fold increase was noted (*p* = 0.003).

Subsequently, we decided to analyze the protein levels of two key enzymes involved in the glucose metabolism pathways: hexokinase 1 (HK1) and glucose-6-phosphate dehydrogenase (G6PD) (Figure 6A). No significant changes were observed in the cortical levels of G6PD (ANOVA *p* = 0.18, F(2, 12) = 1.97, Figure 6B). The level of HK1 had a 0.7-fold decrease in the frontal cortex of the WD (*p* = 0.037) as compared to the control (ANOVA *p* = 0.0004, F(2, 12) = 16.15; Figure 6C). Exposure to moderate intensity training reversed this decline. The level of cortical HK1 significantly increased in the group of exercised WD rats as compared to the WD rats (*p* = 0.0003) and to the CTR group (*p* = 0.03).

In the next step, the authors evaluated the influence of simultaneous exposure to a Western diet and exercise on brain glucose transport. In the frontal cortex, two isoforms of GLUT1 (45 kDa and 55 kDa), GLUT3, and GLUT8 were measured (representative blots Figure 7A). No statistical changes were observed in the GLUTs in frontal cortex of animals fed with a Western diet (GLUT1 45kDa: Kruskal–Wallis test: *p* = 0.28; GLUT1 55 kDa: ANOVA *p* = 0.64, F(2, 12) = 0.45; GLUT8 ANOVA *p* = 0.045, F(2, 12) = 4.06, Figure 7B–E). We reported an almost 2-fold increase in cortical levels of GLUT3 (*p* = 0.04), a major neuronal glucose transporter, in animals fed with a WD and exercised in wheels (WD/EX group), as compared to the WD group (ANOVA *p* = 0.028, F(2, 12) = 4.872; Figure 7D).

Furthermore, the effect of exposure to a Western diet and exercise have been evaluated for the cortical level of MCTs (Figure 8A). No statistical changes were observed in the MCT1 levels (ANOVA *p* = 0.078, F(2, 11) = 3.22, Figure 8B). In response to a Western diet feeding, statistical changes in MCT2 levels were observed (ANOVA *p* < 0.0001, F(2, 12) = 38.79). In the WD group, MCT2 level was reduced as compared to the CTR (*p* = 0.0022) (Figure 7C). The wheel-running reversed this decline: a 1.54-fold increase in MCT2 was observed in the WD/EX rats (*p* < 0.0001). In the exercised rats fed with a WD, a 0.68-fold decrease in MCT4 was observed, as compared to the WD (*p* = 0.005) and to the CTR (*p* = 0.028) (ANOVA *p* = 0.005, F(2, 12) = 8.47; Figure 7D). In the WD/EX group, the MCT5 level was significantly increased as compared to the CTR (*p* = 0.006) and to the WD (*p* = 0.004) (ANOVA *p* = 0.0022, F(2, 9) = 13.0, Figure 7E).

## 4. Discussion

The novelty of the present study is the determination of cerebrocortical proteomic profile in a model of wheel-running exercise, while previous investigations focused primarily on proteomic analysis in various peripheral tissues (e.g., skeletal muscles, liver) of trained animals. Moreover, the proteomic responses of nervous tissue following a high-fat, high-sugar diet and exercise training in female rats have been rarely studied thus far. In the present study, we used an obesogenic diet with human snacks, which is a well-described model of diet-induced obesity in laboratory rodents [29,30]. The potential of physical exercise to decrease body weight, alleviate depression, combat aging, and enhance cognition has been well-supported by animal and human studies. In animal research, exercise regimens vary widely across experiments, which makes it hard to directly compare observed effects. Moreover, some prior studies suggest that voluntary and forced exercises may differentially affect the brain and behavior [31,32]. We employed a wheel-running system (motor driven)—which allowed us to control the training intensity and avoid additional stress associated with commonly used treadmill training with electrical stimuli. We observed that wheel-running training improved some endocrine features related to obesity, i.e., the serum levels of glucose and leptin. However, the basic parameters of the obesogenic phenotype—animal weights, fat and lean body masses (muscles and skeleton)—were similar to the ones observed in the WD rats, which can be due to the gross assessment of these parameters. Previously, it was shown that a 4-week exercise training resulted in sustained weight gain in females [33].

An analysis of the brain proteome revealed that the expression of proteins in the temporal cortex of exercised rats fed with a Western diet was clearly altered, while minor changes were detected as a result of the consumption of a Western diet without wheel-running training. The post hoc analysis of the proteomic data from the three experimental groups revealed significant changes in the expression of seven proteins in the WD rats. Being fed with a Western diet resulted in the upregulation of three proteins: ATP synthase (ATP5j), Alpha-1-inhibitor 3 (A1i3), and Acetyl-CoA acetyltransferase (Acat1). Concurrently, four proteins were downregulated: Acyl-CoA dehydrogenase family member 9 (Acad9), Contactin-associated protein 1 (Cntnap-1), Disks large homolog 4 (Dlg4), and Protein kinase C (PKC). Here, it is worth mentioning that the aim of our study was to obtain the general picture of the effect induced by a combination of a Western diet and wheel-running training. The results of proteomics allow us as the authors to explore the potential baseline reference for further studies on brain changes induced by lifestyle modifications/unhealthy lifestyle. Hence, the validation of such data in the temporal cortex using Western blot would be very interesting but is beyond the scope of the present study. Due to the importance of mitochondrial energy metabolism, we focused our attention on three proteins: ATP5j, Acat1, and Acad9. In the frontal cortex of WD rats, the changes in the protein levels of Acad9, Acat1, and ATP5j are in line with those shown in the analysis of the global proteome. Acad9 promotes dehydrogenase activity toward a broad range of substrates with greater specificity for long-chain unsaturated acyl-CoAs, beta-oxidation of acyl-CoA, and amino acid catabolism. This protein was recently shown to be crucial for oxidative phosphorylation complex I assembly, and it is also considered the main site of oxygen radical production in mitochondria [34]. In the present study, the levels of Acad9 and ATP5j were significantly modulated by a Western diet in the frontal cortex. Six weeks of wheel-running did not affect the changes induced by being fed a Western diet. Namely, the cortical levels of Acad9 and ATP5j in the WD/EX group did not differ from those in the WD group. The results from previous studies indicated that treadmill training improves mitochondrial bioenergetics in the brain cortex and cerebellum [35] and provides mitochondrial protection against oxidative damage in the gastrocnemius muscle [36]. Moreover, the results arising from other experimental studies show that regular, moderate aerobic exercise promotes antioxidant capacity in the brain. In contrast, anaerobic or high-intensity exercise, aerobic-exhausted exercise, or the combination of both types of training could deteriorate the antioxidant response [37,38]. We concluded that the decreased levels of Acad9 may be a result of enhanced oxidative stress as a synergistic effect of exposure to a Western diet and wheel-running training. It is a matter of speculation if the unexpectedly consistent decrease in Acad9 across all studied groups might become a signal for the activation of potent, protective antioxidative mechanisms. However, such conjecture warrants further studies. Moreover, Guo et al. (2013) using a proteomic approach found that ATP5j (as part of the oxidation–phosphorylation pathway) was upregulated in the liver mitochondria of high-fat diet-fed diabetic mice [39]. In the study of Huang et al. (2018), an inactivation of ACAT1 in the myeloid cell lineage improved insulin sensitivity and suppressed a Western diet-induced obesity in mice [40]. Recently, Zhao et al. (2021) showed that a high-fat diet increased the level of ACAT1 in the liver of mice; this effect was normalized in mice fed with a high-fat diet and additionally exercised on treadmill wheels [41]. In this context, the absence of an experimental group with exercised rats fed with a standard diet may be considered a limitation of the present study. We have focused on the changes in proteins and pathways that can serve as a reference for better understanding and a reason to expand exercise usage as a therapeutic strategy for CNS diseases. With this purpose in mind, a quantitative proteomic approach has been undertaken to identify the protein expression changes in the rat brain as the effect of simultaneous exposure to a Western diet and wheel-running. It is worth mentioning here that we deliberately used the temporal cortex for the brain proteome for several reasons. The first one is linked to the fact that several recent works outlined that Western diet/high-fat diet-induced changes extend beyond the hypothalamus to affect areas directly related to cognition, such as the frontal and parietal cortex [18]. Secondly, some aspects of this Western diet model have been described previously [25], and the proteomic approach used in the present study may add substantial pieces to the puzzle if conducted under the same experimental conditions. Finally, in order to reduce the number of animals in the experiment, in accordance with the 3R principle, we conducted proteome and Western blot studies on the same cohort of rats.

Abnormal glucose metabolism and transport underlie disorders of the brain and the whole organism. It is well known that changes in glucose metabolism are associated with the etiology of neurodegenerative diseases [42,43,44] and depression [45]. Referring to the above-mentioned disturbances of the brain energy metabolism induced by a Western diet and functional annotation data of a protein list identified by LC-MS/MS, we have additionally analyzed the cortical levels of hexokinase-1 (HK1), glucose-6-phosphate dehydrogenase (G6PD), as well as glucose (GLUT1, GLUT3, and GLUT8) and monocarboxylate (MCT1, MCT2, MCT4, and MCT5) transporters. While a significant decrease in the cortical levels of HK1 and MCT2 were found after exposure to a Western diet, the other studied proteins remain unchanged. Of special interest may be a lack of significant changes in the levels of G6PD, which is a key regulatory enzyme in the pentose phosphate pathway (PPP). G6PD expression and activity are critical for maintaining the proper redox cellular status [46], and its retained activity in the WD group may be—hypothetically—a protective mechanism. Previously, a significant decrease in the mRNA expression of HK1, glucokinase, and pyruvate kinase was observed in the livers of mice fed with a high-fat diet (HFD) [47]. In study by Pierre et al. 2007 [48], the authors showed that MCT2 expression was neither increased in an extract containing cingulate and motor cortex nor in the cerebellum of mice fed with a HFD compared to mice fed with a standard diet. By contrast, it was significantly enhanced in extracts composed of piriform, insular and entorhinal cortex, and hippocampus. However, there are no studies evaluating the effect of regular exercise on unhealthy, diet-induced disturbances of cerebral energy metabolism. Similar to our results of proteomic analysis, exposure to a diet and wheel-running changed the levels of most of the proteins studied. In the WD/EX rats, the cortical levels of HK1, GLUT3, MCT2, and MCT5 increased, and the level of MCT4 decreased. It is worth emphasizing here that wheel-running increased the cortical levels of two proteins: HK1 and MCT2. We may hypothesize that physical activity may increase the availability of glucose for neurons by increasing glucose transport into the cells through MCT2. Concomitantly, the augmented glucose phosphorylation by hexokinase to glucose-6-phosphate, which is a substrate not only for glycolysis but also possessing extreme antioxidative importance via the pentose phosphate pathway, as it enhances the protective intracellular antioxidant mechanisms [49]. Regular physical activity has a positive effect on the functions of the CNS; however, little is known about the exact mechanisms by which the skeletal muscle activity and whole body energy challenge exert long-lasting beneficial changes in the CNS. It has been recently demonstrated that physical activity reduces the development of age-related skeletal muscle insulin resistance by enhancing the expression of GLUT1 and GLUT4 [50], whereas acute exercise increased GLUT1 expression in the rat cerebral cortex and both MCT1 and MCT2 in the cortex and hippocampus [51]. It has been proposed that intermittent energy challenges such as regular aerobic activity or fasting may exert positive effects on brain function acting through similar mechanisms [14]. Energy challenges stimulate pathways associated with the usage of alternative energy sources such as monocarboxylates [52]. Brain MCTs have been proposed to play a key role in coupling neuronal activity and lactate transport through a mechanism referred to as the astrocyte–neuron lactate shuttle (ANLS). Given the key role of the ANLS in mediating lactate transport for memory formation, it is feasible that exercise-induced increases in monocarboxylate and MCT proteins in the brain may benefit brain functions, including cognitive ones [51].

It is widely postulated that unhealthy, obesogenic diets and exercise training have opposite impacts on health, especially on the brain. We concluded that proteins that are differentially regulated by Western diet and wheel-running are particularly promising candidate proteins that contribute to exercise-induced health benefits. This rationale is supported by showing the downregulation of the cortical levels of Acad9, HK1, and MCT2, which in our opinion provides strong evidence for the disruption of brain energy metabolism as a response to a hypercaloric diet. Further studies are warranted to investigate other brain metabolism-related proteins as suggested by the results of proteome analysis.

## Figures and Tables

**Figure 1 nutrients-13-04242-f001:**
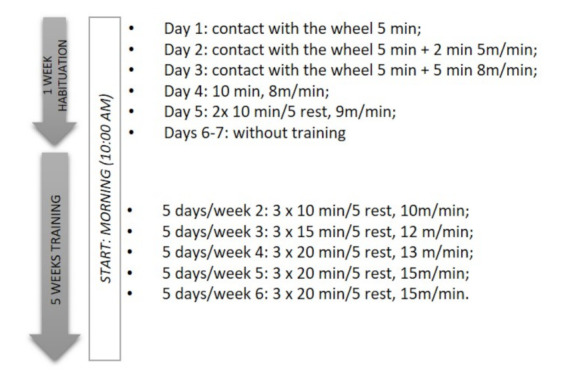
The schedule shows the wheel-running training protocol. The rats were exercised 5 days/week for 6 weeks, of which the first week was an introduction week (the habituation phase). The speed, time, and number of sessions by day are described.

**Figure 2 nutrients-13-04242-f002:**
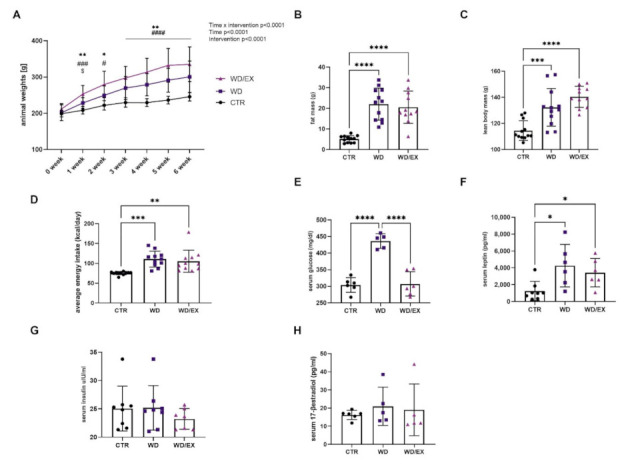
Animal weights, body composition, energy intake and serum biochemical parameters of control rats (CTR, *n* = 12), rats fed with western diet (WD, *n* = 12), and exercised rats fed with western diet (WD/EX, *n* = 11). The data shown in panels A-F were collected and analyzed from three experimental groups. Data are expressed as means ± standard deviation (SD). (**A**) Body weight of rats recorded weekly during the 6-week feeding period. Two-way ANOVA, Tukey’s multiple comparisons test: Significant differences are indicated by *** *p* < 0.001, ** *p* < 0.01, * *p* < 0.05: CTR vs. WD; # *p* < 0.05, #### *p* < 0.0001, ### *p* < 0.001: CTR vs. WD/EX; $*p* < 0.05: WD vs. WD/EX; (**B**) Fat mass, (**C**) Lean body mass expressed in grams. One-way ANOVA; Tukey’s multiple comparisons test: Significant differences are indicated by **** *p* < 0.0001, *** *p* < 0.001; (**D**) Average energy intake calculate as per rat/kcal. One-way ANOVA, Tukey’s multiple comparisons test: Significant differences are indicated by *** *p* < 0.001, ** *p* < 0.01; (**E**) Serum glucose expressed in mg/dL. One-way ANOVA, Tukey’s multiple comparisons test: Significant differences are indicated by **** *p* < 0.0001; (**F**) Serum leptin levels expressed in pg/mL. One-way ANOVA, Tukey’s multiple comparisons test: Significant differences are indicated by * *p* < 0.05; (**G**) Serum insulin expressed in µlU/mL; and (**H**) Serum 17-β estradiol expressed in pg/mL.

**Figure 3 nutrients-13-04242-f003:**
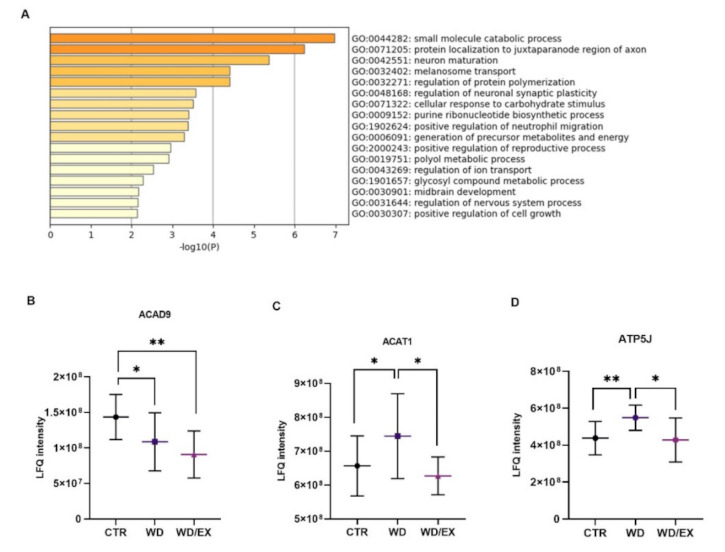
(**A**) Summary of significantly changed biological processes for proteins, which were significantly different in the multiple sample ANOVA analysis (CTR vs. WD vs. WD/EX) (Metascape); (**B**–**D**) Three representative proteins showing changes in the expression mitochondrial proteins in the cerebrocortical samples after exposition to the Western diet and wheel-running. The data shown in panels (**B**–**D**) were collected and analyzed from three experimental groups: control rats fed with standard diet (CTR, *n* = 12), rats fed with Western diet (WD, *n* = 11), and exercised rats fed with Western diet (WD/EX, *n* = 7); LFQ intensity—Label-Free Quantitation calculated by MaxLFQ algorithm to determine the relative amount of proteins in more biological samples. Data are expressed as means ± SD; One-way ANOVA, Fisher post hoc test: Significant differences are indicated by * *p* < 0.05, ** *p* < 0.01.

**Figure 4 nutrients-13-04242-f004:**
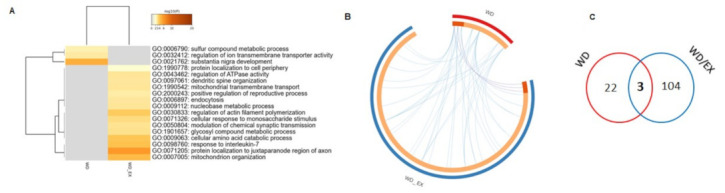
(**A**) Enriched terms (GO: BP) identified for proteins that were significantly different in the WD and WD/EX groups (both in comparison to CTR), filtered, and hierarchically clustered (Metascape); (**B**) The circos plot shows how genes from the input gene lists overlap. On the outside, each arc represents the identity of each gene list (red: WD; blue: WD/EX). On the inside, each arc represents a gene list, where each gene has a spot on the arc. Dark orange color represents the genes that appear in multiple lists and light orange color represents genes that are unique to that gene list. Purple lines link the same gene that are shared by multiple gene lists. Blue lines link the different genes where they fall into the same ontology term (Metascape); (**C**) Venn diagram of altered proteins.

**Figure 5 nutrients-13-04242-f005:**
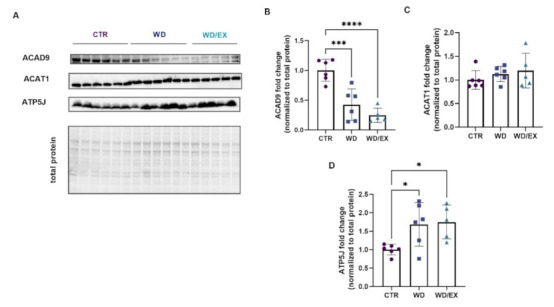
Mitochondrial proteins involved in glucose metabolism pathways in the frontal cortex. Western blot images (**A**) and quantification of (**B**) Acyl-CoA dehydrogenase family member 9—Acad9, (**C**) Acetyl-CoA acetyltransferase, mitochondrial—Acat1, and (**D**) ATP synthase-coupling factor 6, mitochondrial—ATP5j measured in the frontal cortex of rats fed for 6 weeks with standard diet (control, CTR, *n* = 6) or Western diet (WD, *n* = 6), or exercised and fed with Western diet (WD/EX, *n* = 5) and normalized to total protein levels (representative loading control is shown corresponding to the Acad9). Data are expressed as mean values ± SD; one-way ANOVA, Tukey’s test: significant differences are indicated by * *p* < 0.05, *** *p* < 0.001, **** *p* < 0.0001.

**Figure 6 nutrients-13-04242-f006:**
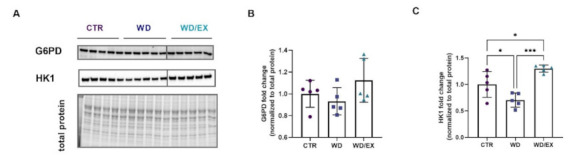
Enzymes involved in principal glucose metabolism pathways in the frontal cortex. Western blot images (**A**) and quantification of (**B**) glucose-6-phosphate dehydrogenase (G6PD) and (**C**) hexokinase-1 (HK1) measured in the frontal cortex of rats fed for 6 weeks with standard diet (control, CTR, *n* = 6) or Western diet (WD, *n* = 6), or exercised and fed with Western diet (WD/EX, *n* = 5) and normalized to total protein levels (representative loading control is shown corresponding to the G6PD). One-way ANOVA followed by Tukey’s test was used; mean values ± SD are shown on each graph. * *p* < 0.05, *** *p* < 0.001.

**Figure 7 nutrients-13-04242-f007:**
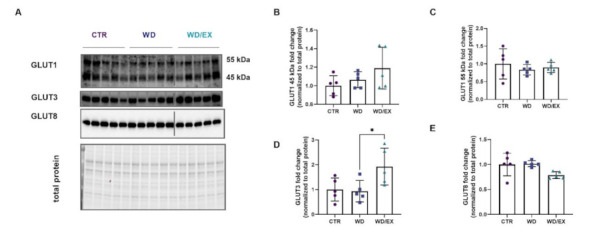
Glucose transporters (GLUTs) expressed in the frontal cortex. Western blot images (**A**) and quantification of (**B**) GLUT1 45kDa, (**C**) GLUT1 55 kDa, (**D**) GLUT3, and (**E**) GLUT8 measured in the frontal cortex of rats fed for 6 weeks with standard diet (control, CTR, *n* = 6) or Western diet (WD, *n* = 6), or exercised and fed with Western diet (WD/EX, *n* = 5), and normalized to total protein levels (representative loading control is shown corresponding to the GLUT1). One-way ANOVA followed by Tukey’s test was used; mean values ± SD are shown on each graph. * *p* < 0.05.

**Figure 8 nutrients-13-04242-f008:**
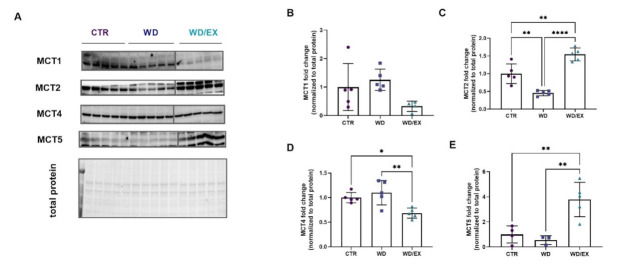
Monocarboxylate transporters (MCTs) expressed in the frontal cortex. Western blot images (**A**) and quantification of (**B**) MCT1, (**C**) MCT2, (**D**) MCT4, and (**E**) MCT5 measured in the frontal cortex of rats fed for 6 weeks with standard diet (control, CTR, *n* = 6) or Western diet (WD, *n* = 6), or exercised and fed with Western diet (WD/EX, *n* = 5) and normalized to total protein levels (representative loading control is shown corresponding to the MCT2). Data are expressed as mean values ± SD; one-way ANOVA, Tukey’s test: Significant differences are indicated by * *p* < 0.05, ** *p* < 0.01, **** *p* < 0.0001.

## Data Availability

The data that supports the findings of this study are available on request from the corresponding author.

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
