# Peer review of "Effects of Simultaneous Exposure to a Western Diet and Wheel-Running Training on Brain Energy Metabolism in Female Rats"

_nutrients, 2021, doi:10.3390/nu13124242_

Round 1
Reviewer 1 Report
This manuscript provides evidence for alterations to proteome of the temporal and frontal cortex as a result of western diets and exercised in female rats. Using mass spectrometry and western blot analyses, the authors found that western diets reduced the levels of Acad9, regardless of exercise regiment. They also found that western diets reduce levels of HK1 and MCT2, effects that were rescued by exercise. These findings provide insight into the neurobiological effects of western diets on the brain's proteome.
However, some of the authors conclusions are not supported by the data. In addition, the overall goal of the manuscript is unclear, as the authors jump between emphasizing western diets and/or exercise. The rationale to use two different methods on two different brain regions is unclear, as are the links between these two parts of the manuscript. Although the N for mass spectrometry experiments is good, it is unclear why a lower N was used in the western blots and there seems to be a large amount of noise in the western blot data. Overall, I have some concerns regarding the reliability of analyses and their interpretation. My specific comments and concerns are outlined below.
Abstract:
- Several of the findings presented as significant in the abstract are not supported by the data, including the highlighted alterations to G6PD and MCT4. I would also argue that exercise did not "normalize" HK1 and MCT2 levels, as they remained significantly different from controls. I think this finding in particular should be further expanded on in the discussion.
Introduction:
- Please add references to support the points made in lines 83-88.
Methods:
- Please describe how energy intake was measured.
- The authors mention that the hippocampus was also analyzed by western blot, but those results are not presented, nor are the methods for hippocampal collection described. Is this a mistake or were these results omitted from the manuscript?
- Please provide additional details on how total protein was quantified.
- It is unclear how many replicates were performed for the western blot analyses. Please include these details in the main text.
- The statistical analyses performed for the mass spectrometry analyses require more detail. In particular, it is unclear which post hoc tests were performed to identify genes that were specific to the two treatment groups.
Results:
- Please include graphs of serum insulin and estradiol, even if non-significant.
- I am uncertain of the value of presenting the gene ontology results for the 80 proteins identified across both treatment groups. This approach does not seem particularly informative, as it is confounded by the two treatments. I would suggest the authors present gene ontology for the two sets of genes. Furthermore, it is unclear whether the differential genes that were "specific" to each treatment groups were only significantly different in the group of interest. It would be important to show which genes are specific to each treatment and which genes overlapped (a venn diagram would be an easy and intuitive way to present some of these results). That said, I think the results of the mass spec experiment are interesting and important, hence my suggestions to improve their interpretability.
- Regarding the mass spec results again, it would be useful to provide a table summarizing the main findings, with p-values, mean levels, fold change, etc. for the top differential proteins.
- Additional rationale is needed to justify why the authors selected these specific proteins for follow-up analyses in a different brain region. As it stands, the link between part 1 and part 2 is rather unclear.
- It is interesting that the differences in protein levels do not persist across the mass spec and western blot analyses. Although this could be due to differences in brain regions, it would be important to know how the authors selected the subset of animals for this analysis. It think it would be important to show a direct comparison of protein levels from the mass spec and western blot analyses for those animals, as the N is considerably smaller.
- I will note this here, as it is part of the main conclusions in several parts of the manuscript. The analysis of G6PD did not identify any differences between groups and therefore should be discussed as such. Furthermore, the presentation of MCT4 results is rather misleading in some parts of the manuscript, as it is only significantly different in the WD/EX group.
- There appear to be some missing results in from lines 339-346, as MCT1 and MCT5 were not presented.
Discussion:
- Overall, I think the discussion could be better organized and provide a better contextualization of the findings. There are several parts that read more as rationale for the experiments, which would be better suited earlier in the manuscript (e.g. lines 414-426). There is also some confusion as to the main goal of the analyses, as the authors seem to jump between WD and exercise as driving factors for the differences they observed.
- The authors should discuss the potential differences arising from analyzing two different brain regions with two different methods. As it stands, I feel like these should be two different papers altogether, as the authors do not link these two sets of results at all.
- There appears to be some discussion lacking around the use of female animals only. Although I completely agree with the authors that the analysis of female animals is highly relevant and of great importance to the field, there should be some discussion of whether they think these effects would also be present in males.
- Lines 401-411. This part of the discussion is not supported by the data presented by the authors. I would argue that their findings suggest that exercise does not rescue the effects of WD on the brain and that WD is the main driver of changes in Acad9, based on the western blot and mass spec results. The authors should revise this part of the discussion and make clear which parts are in reference to their own results, versus those from the literature.
- Are any of the differential proteins identified in the present study found in genome-wide association studies of psychiatric disease? The authors should check the lists published by the Psychiatric Genomics Consortium, as this would help strengthen their case for the relevance of this work to mental illnesses.
- Line 441 - missing a reference for the Pierre et al study.
Figures and tables
- Figure S1 - this figure would be more interpretable if the samples were not clustered and instead grouped together by treatment. There should also be a legend for the colors of the heatmap and the sample information. As it stands, the directionality of changes is unclear. It is also unclear whether the colors represent base protein levels or Z-scores across rows (which would be more informative).
- Table S1 - please include the adjusted p-value. It is also alarming that these values do not match those in figure 3B-D.
- Figure 3 legend - I am confused by the mention of chronic stress. Please define LFQ intensity.
- Figures 5A and 6A - why is there a line in the western blot? Are these two different gels?
- For the western blots, it would be helpful if the authors included a box showing how they selected the bands for analysis. In several instances, there are multiple bands and it is unclear which was analyzed. The authors should also mention the fact that they may be analyzing multiple isoforms in the discussion.
Minor points:
- Findings should be not be described as "statistically significant" or "statistical changes" on lines 176, 181, 209, 326.
- I am not certain what the authors mean by the "non-obvious nature of the results" on line 467 - does this mean they do not know how to interpret them? My apologies, but the fact that this is the last sentences leaves me with some doubts about the conclusions from this study. I think the authors could have a stronger takeaway and conclusion.
Reviewer 2 Report
In this manuscript, Marta Nowacka-Chmielewska and colleagues investigated the impact of a western diet and wheel running training on metabolic outcomes including cerebrocortical proteomic profile using a rat model. The idea of the study and findings were of interest; however, the organization of results and discussion needs to be improved. There was substantial text recycling in the methods such as the dietary protocol section, which is not properly conducted.
Specifically:
- The results section could be improved as the comparisons were not clearly stated, which could be confusing to readers.
- The abbreviation WD seemed to be used for both western diet and the treatment group, which was confusing.
- The usage of first-person and third-person pronouns was not consistent. Please revise accordingly.
- There was limited background information/discussion about identified proteins. The discussion needs to be elaborated by providing more information from previous studies.
- L 33-35 – Abbreviations should not be used in the abstract.
- L 61-62 – Please check through the manuscript with respect to abbreviations. Please avoid both the overuse and underuse of abbreviations. If you abbreviate a term, use the abbreviation at least three times in a paper. Abbreviations should be written out in full on first use.
- L 95 – Was a power analysis used to confirm the minimum number of animals? If so, please provide parameters used for the power analysis.
- L99-100; L121 – “ad libitum” should be italicized.
- L 112-123; L 145-158; L 162-167 – Although a reference has been provided, there was substantial text recycling in the method section (simply cut and paste from the previous paper), which is not appropriate.
- L 120 – Please provide information about energy content in fructose solution. How much did rats consume daily? How did fructose consumption contribute to calorie intake?
- L 133-134 – Daily.
- L 145 – Were the animals fasted before tissue collection?
- L214-218; Figure 2A; L 370-373 – How about multiple comparisons in body weight between groups at each week?
- L 256-260 – Should it be FDR p value (q value)?
- L 269 – It was not clear to readers which group the WD/EX group was compared with.
- L 280 – the chronic stress?
- L 297-298 – Please revise the statement to make it clearer.
- L 343-344 – “compared with CTR rats”.
- L 401-403 – It was not clear whether the statement was about previous studies or the current study.
